

# A global comparison of community-based responses to natural hazards

Barbara Paterson[1], Anthony Charles[1]

[1]School of the Environment, Saint Mary's University, 923 Robie Street, Halifax, Nova Scotia, Canada, B3H 3C3

*Correspondence to*: Barbara Paterson (Barbara.paterson@smu.ca)

**Abstract.** Community-based disaster preparedness is an important component of disaster management. Knowledge of interventions that communities utilise in response to hazards is important to develop local-level capacity and increase community resilience. This paper systematically examines empirical information about local level responses to hazards based on peer reviewed, published case studies. We developed a data set based on 188 articles providing information from 318 communities from all regions of the world. We classified response examples to address four key questions: (i) What kinds of responses are used by communities all over the world? (ii) Do communities in different parts of the world use different kinds of responses? (iii) Are communities using hazard-specific responses? (iv) Are communities using a multi-hazard approach? We found that within an extensive literature on hazards, there is relatively little empirical information about community-based responses to hazards. Across the world, responses aiming at securing basic human needs are the most frequently reported kinds of responses. Although the notion of community-based disaster preparedness is gaining importance, very few examples of responses that draw on the social fabric of communities are reported. Specific regions of the world are lacking in their use of certain hazard responses classes. Although an all-hazards approach for disaster preparedness is increasingly recommended, there is a lack of multi-hazard response approaches on the local level.
.

## 1 Introduction

Natural disasters such as floods, storms and extreme weather events put many people at risk, especially in coastal regions. Between 1994 and 2013, on average 218 million people were affected annually through losing their homes or livelihoods due to natural disasters, and 1.35 million number of people died over this period (Creed, 2015).

The intensity and frequency of hazardous events is increasing with climate change leading to growing numbers of annual disasters all over the world (IPCC 2014). The growing body of literature on disaster risk reduction, community resilience and adaptive capacity attests to the increased focus on finding ways through which people can prepare for this, to reduce the impact of hazards and to increase the ability of local communities to cope with the consequences of hazardous events.



The systematic global approach to disaster risk reduction (DRR), which was initiated through the Hyogo framework (2005-2015), has triggered much progress in disaster management in many parts of the world. However, the ongoing need for significant improvement of disaster mitigation measures, particularly in less developed countries, underpins the targets of the subsequently adopted Sendai framework for disaster risk reduction (2015-2030). The 20 year review (1993-2013) of disaster

impacts conducted by the Center for Research on the epidemiology of disasters (CREED) comes to similar conclusions.

The latter report also posits that more progress is needed in understanding how and why people, especially at the local level, are affected by disasters – so as to enable DRR strategies that are based on evidence rather than assumptions (CREED, 2015). The need for this is reinforced by a review by Gall et al. (2015), which reveals that most disaster risk research is

academic and provides only limited evidence for policy improvement.

Although local communities have little influence on the hazard itself, communities have the potential to function and adapt successfully in the aftermath of disasters (Norris et al. 2008). Community-based disaster preparedness is increasingly considered an important component of disaster management with the potential to increase community resilience through

local-level capacity building (Allen, 2006). The actual assessment of community resilience remains a key challenge, however (Cutter et al. 2008), despite ongoing research to understand the nature of resilience to hazards more broadly.

One piece of the puzzle to improve community resilience to hazards is better knowledge of the kind of interventions that communities are already utilising in order to respond to hazards. Thus, this study aims to shed light on how people in

communities are responding to hazards. This is done by classifying the kinds of responses used, into a typology providing an opportunity to understand the diverse character of community-based responses. A typology of responses also facilitates comparison of how people respond in different places or to different hazards.

**1.1 Community-based responses to hazards**

In the context of this paper, a 'response' is any intervention or activity that addresses the likelihood or consequences of

hazards and thus aims to increase community resilience. Although contested, the term 'community' is defined in a place-based manner, to refer to a group of individuals and households living at the same location, i.e. a specific local-level geographic area, such as a village or group of villages, a municipality, a small city or a neighbourhood.

Scale plays a major role in the consideration of resilience to hazards. While physical processes, such as climate change, that

drive hazards and cause increased frequency and intensity of disasters, are global, the impacts are most acutely felt at the local level.



This paper systematically examines empirical information about local level responses to hazards based on peer reviewed, published case studies. Through this analysis and the typology approach, we are aiming to address four key questions:

1. What kinds of responses are used by communities all over the world?
2. Do communities in different parts of the world use different kinds of responses?
3. Are communities using hazard-specific responses?
4. Are communities using a multi-hazard approach?

## 2. Methods

We conducted a systematic review of case studies published in English language peer-reviewed literature. Each article describes one or more interventions that were implemented at the local level in places where people have been directly experiencing the impacts of hazards arising in many forms (e.g., floods, storms or earthquakes). We searched 9 different databases across multiple disciplines which required different search terms or keywords depending on the database used. Most search strings followed the format "(hazard OR storm OR hurricane OR cyclone OR earthquake OR flood …) AND
communit*". Bibliographies of relevant papers were screened.

The search resulted in 1671 articles for full text analysis. The aim of the literature search was to find studies that were conducted after people had experienced the negative effects of a hazard event, and which provided information on how people responded to or coped with this event, whether immediately following the event or in the longer term afterwards.
Therefore we only included articles that reported on (i) a hazard event or trend that has caused loss of life or health impacts or damage or loss to property, infrastructure, livelihoods and environmental resources; and (ii) a local community, as defined above. We excluded articles that (1) did not cite a specific event or geographic location, (2) described only potential but not actual actions, (3) provided assessments of risks, impact or vulnerability but no response actions, or (4) described results from modelling exercises. A total of 188 articles (11% of 1671) matched our criteria and were selected for content analysis
(Table 1).

The date range of articles in our data set is 1982-2014, however the large majority of case studies fall within the period between 2001 and 2014. From the 188 articles, basic information was extracted about the community (name of the community, longitude, latitude, country), the hazard event/disaster (type of hazard, year, description or context, type of
exposure, a description of the interventions/response actions and the aim of the intervention (to modify the event, to modify people's vulnerability, to modify loss).





We coded the descriptions of response actions using a grounded theory approach to label the sentences or paragraphs discussing a particular response action, which resulted in 899 response codes. Similar responses were grouped first into 15 types and then into five classes according to the purpose of the actions or interventions that were taken. Because we used a qualitative approach, the number of types within each class varies.

In order to compare the frequencies of response classes between different types of natural hazards, we grouped the hazards into four types based on the EM-DAT classification of natural disasters over the period 1994-2014 (http://www.emdat.be/classification). EM-DAT classifies disasters into five groups: Geophysical, Meteorological, Hydrological, Climatological and Biological. Although not all hazard events in our data set are of disaster proportions, we

adopted the same classification for the hazardous events, and here use the labels Earth, Weather, Water, and Climate for the four hazard types examined (excluding biological hazards here). Geographic analyses of reported responses is based on the United Nations' scheme of geographic regions.

A number of possible biases in the methods should be noted. First, because the sample of response examples has been taken

from the peer reviewed English literature, it could be subject to a language bias. However, English is considered the international language of science (Tardy 2004; Kirchik et al. 2012) and although this notion may be problematic, it is not the subject of this paper. Second, it is possible that the response examples extracted from the articles represent the priorities of the authors rather than the realities of what took place in local communities. However, use of secondary data allows compilation of a global data set of local response actions, and the results obtained from that match findings of more specific

research, which suggests that the data obtained do lead to a comprehensive and reliable global picture of community-based responses to hazards. Thus a data set based on peer reviewed English language articles is at the very least a useful starting point for development of a global perspective on community based responses to hazards.

## 3. Results & Discussion

Our data set includes information for 318 communities from all regions of the world, reflecting the fact that some of the 188

articles refer to more than one community each. Asia has the highest research coverage, both in terms of number of articles (table 2) and in terms of the number of communities (162, i.e. over half of the total) for which data is reported. Africa has the lowest research coverage, with only 5% of articles, and reporting on a total of 12 communities. The most studied countries in their respective regions are USA (86% of Northern America) followed by UK (45% of Europe) and Australia (39% of Oceania). Since, as noted above, only a small proportion of reviewed case studies (11% of 1671) provide empirical

information on local responses to hazards, very little data is available across all regions (which corresponds to similar findings by Gall et al. 2015) but in particular for the African region.





### 3.1 What kinds of responses are used by communities? A typology of response actions

The results of the analysis of response examples into types and then into major classes are shown in Fig. 1. The resulting classes are: (1) Individual and material wellbeing; (2) Relational wellbeing; (3) Awareness of hazards and risks; (4) Guidance and governance; and (5) Infrastructure. These are described below, together with a sample of hazard response examples for each (organized by response type).

### 3.1.1 Individual and Material Wellbeing

Responses grouped in this class are about securing the necessities of life. This class comprises seven different types of actions that emerged from the qualitative analysis (Fig. 2). Actions to ensure the availability of *food and shelter* include examples such as the planting and storing of food crops for emergency rations during cyclone season in Fiji and Vanuatu (McNamara and Prasad 2014) or the use of local schools as temporary shelter after a tsunami in Malasia (Bird et al. 2007). To repair or replace damaged property or restore people's health requires *financial* means and response actions in this category include the use of collective savings to recover from wild fire damage in the Philippines (Dodman et al. 2010) and the purchasing of flood insurance in Italy (Miceli et al. 2008).

Actions which ensure a continued source of income after a hazardous event are grouped as *livelihood adaptations.* Responses of this type are mostly agricultural ranging from adapting planting schedules to reduce loss from floods in South Africa (Vermaak and van Niekerk 2004 ) and Malawi (Chidanti-Malunga 2011 ) to integrating duck and fish farming on rice paddies in Vietnam to take advantage of the flood season (Nguyen and James 2013). People *protect belongings* from damage by stacking or elevating furniture and other possessions during floods in Bangladesh (Marfai et al. 2008) or prevent damage to important assets such as fish traps by moving them inland during the hurricane season in Anguilla (Forster et al 2014). People change their emotional state or protect their mental health through *psychological* responses. Examples include framing the event as the will of a higher force (Mehta 2001) or the shifting from despair to enthusiasm in the process of creating value through deconstructing buildings destroyed during Hurricane Katrina (Denhart 2009). *Mobility* responses reduce hazard exposure by physically removing people from areas that are at risk. Examples include both permanent or seasonal migration away (e.g. Penning-Rowsell et al. 2013; Kartiki 2011; Sudmeier-Rieux 2012 ) as well as evacuations (Esteban et al. 2013). Search and rescue actions focus on saving lives such as after the 2000 floods in the Limpopo province of South Africa (Vermaak and van Niekerk 2004) or after the 2009 Tsunami in American Samoa (Rumbach and Foley 2014). One could argue that as evacuations aim to protect people from physical harm they have the same purpose as search and rescue. Moreover, since evacuations are also part of the immediate emergency response activities, these kinds of responses could be grouped together. However, we feel that the salient property that sets evacuations apart from other disaster responses is the physical removal of people from the hazard area. We thus grouped evacuation with migration under the heading *Mobility.* Interventions that focus on moving people out of harm's way or resettling victims in a new location





after a disaster, can negatively affect the social relations and identities among community members. In some cases, these potential negative effects are addressed by moving the entire community. These kinds of relocations, although technically referring to the movement of people, were grouped not in *Mobility* but within the class *relational wellbeing*.

### 3.1.2 Relational wellbeing

Responses focused on relational wellbeing draw on the fabric of the community for preparedness, for protection during a hazardous event or for the recovery afterwards. These responses typically involve collective action and interactions among community members, e.g., utilising kinship networks or formal community networks. In some cases, the fabric of the community is strengthened as a result of the disaster, e.g. in New Zealand where people "developed a real sense of community and doing things together" by supporting others who had lost family members in an earthquake disaster (Gawith
2013, see also Zahari and Ariffin 2013).

### 3.1.3 Awareness of hazards and risks

This class of responses includes two types. The type *information and data* includes actions to gather and communicate information about the threat conditions. Examples of this type range from the monitoring of an approaching cyclone via television, radio and internet (Anderson-Berry and King 2005) to the use of automatic weather stations (Gupta 2007) and
closed circuit television (Lewis 2013) to monitor weather and flood conditions.

Under *Knowledge and learning* we grouped activities with the purpose of increasing response capacity based on improved understanding of the hazard. Examples of this type include raising tsunami awareness and developing a tsunami culture in communities in Japan and in the Philippines (Esteban et al. 2013). In Iceland, residents acquired specialist knowledge about cultivating ash-impacted soil after a volcanic eruption (Bird and Gisladottir 2012). Because past experience can increase
awareness, some communities in Spain have evoked small-scale flood disturbances to avoid large-scale disturbance and collapse (Gomez-Baggethun et al. 2012).

### 3.1.4 Guidance and governance

This class comprises three types of responses pertaining to decision making. The type *governance* includes actions that handle, direct, govern, or control aspects of human hazard interaction ranging from land use planning regulations to reduce
risk of wild fires in Australia (Buxton et al. 2011) to earthquake building codes in Pakistan (Ainuddin et al. 2014) to a local government policy in USA requiring that local residents raise their houses or risk that the entire community loses access to insurance (Colten et al. 2008). In Taiwan a community is officially designated as a "driftwood art area" in order to boost local artists as part of recovery from flooding (Wang et al. 2013). *Planning* responses include examples from Pakistan, where tents, blankets and emergency food rations had been prepared by village based organisations (Ehsan-ul-Haq 2007) and
examples from the UK such as the distribution of sandbags by town and parish councils (Andrew and Knight 2014).





### 3.1.5 Infrastructure

Responses that focus on infrastructure for physical hazard defense are of two types: *hard protection* based on engineering efforts, or utilizing ecological properties for *green protection*. Examples of *hard protection* range from tying down houses and using safety features, such as hurricane shutters, clinching and rafter anchorage in Saint Kitts and Nevis (Hobson 2003) to flood defences such as sandbags, dykes or breakwaters (Esteban et al. 2013) or the flood proofing of high buildings by placing important functions on higher floors (Elnashai et al. 2012). *Green protection* measures include the re-naturalisation of rivers in Italy to avoid floods (Scolobig et al 2008), the replanting of deforested hillsides to prevent landslides in Uganda (Jenkins et al 2013) and replanting of dune vegetation in New Zealand to reduce coastal erosion (Blackett and Hume 2007) the replanting of native trees, grasses and mangroves to prevent erosion and to create natural barrier against storm surges or tsunami impacts in Thailand (Calgaro and Lloyd 2008; Barbier 2006) and Sri Lanka (Porteous 2008).

We found that in our dataset, some 40% of communities employed interventions from the class *Individual & Material Wellbeing* (Fig. 2). The second largest class is *Guidance* with 20% of communities using responses from that class. The classes Awareness (12%), Relational Wellbeing (14%) and Infrastructure (14%) are reported less frequently. Some 56% of communities (177 out of a total of 318) utilise responses from more than one class.

### 3.2 Do communities in different parts of the world use different kinds of responses?

We found that, with the exception of Europe, all regions of the world have the class *Individual and Material Wellbeing* as the most frequently reported (table 3). This result is not surprising as these responses focus on securing basic needs for people, which are of course of immediate importance, and reflect the priorities of disaster response agencies. The latter tend to focus on the physical and economic aspects of vulnerability (Heijmans 2004, p.115), which may further explain why there is such a strong focus on these kinds of responses in the literature.

Almost all of the African response examples in our data set are associated with *Individual and Material Wellbeing* responses, which may reflect how the physical and economic vulnerability of communities in this region increases the severity of hazard impact, such that securing basic needs is a key focus in the aftermath of any hazardous event. As discussed earlier, there is also a possibility, with the methodology used, that this result could merely reflect a tendency for authors who write about African hazards and disasters to focus on *Individual and Material Wellbeing* responses, for some reason.

Between 1995 and 2015 (the period covered best in our data set), six of the 10 countries with the highest proportion of affected people (i.e. people requiring immediate assistance, including displaced or evacuated people) over the total population, were in Africa (Creed 2015). This fact supports the idea that hazard impacts are severe in Africa. Further, of the 12 communities in our data set, eight are reported to experience climate hazards. Since climate can directly affect agricultural production, and agriculture is Africa's largest economic sector (and is of a small-scale nature), climate hazards directly affect people's livelihoods. This may further explain why there is such strong focus on securing basic needs as a



response to these hazards. Indeed, many of the response examples that fall into the class *Individual and Material Wellbeing* are about livelihood adaptations in the face of drought, extreme weather events or floods.

The other two regions with a high proportion of communities utilising *Individual and Material Wellbeing* responses are Asia

(75%) and Oceania (72%). High population densities in disaster-prone regions in Asia put people at high risk from natural hazards. Indeed, disaster statistics show that this region endured the largest part of global disaster impacts between 1994 and 2013 in terms the total numbers of people killed and affected (Creed 2015). Similarly, 12 of the 36 Oceania communities in our data set are situated in Small Island Developing States (SIDS). Being mostly dependent on agriculture, fisheries or tourism, SIDS are highly vulnerable to natural disasters (Creed 2015), which typically affect the entire population (Méheux

et al. 2006; Barnett and Waters 2016; Robinson 2017). Again, it is not surprising that many communities in these regions employ hazard responses that are aimed at basic needs.

In Europe, on the other hand, the most frequently reported responses (60% of local communities in the data set) are from the class *Guidance and Governance*. This result stands out compared with all other regions and raises the question of why this

class of responses would be preferred, only in Europe, over actions that focus on peoples' basic needs (as everywhere else).

One possible explanation is that this particular result reflects choices by the writers of articles, based on assumptions held that impacts from hazards in Europe do not affect people's basic needs (such as food, shelter and livelihood) in as serious a way, so that securing those basic needs is not a priority. This could be a bias in the results if, for example, Europe is more

likely thought of as a source of disaster aid than as a disaster location. Ironically, however, findings by the European Environmental Agency show that the number and impacts of natural hazards have increased in Europe in the period 1998-2009 having caused 100 000 deaths, affected more than 11 million people and led to economic losses of about EUR 150 billion (EEA 2011).

Another explanation lies in the reality that the policy focus in Europe has been on more integrated policies, in particular regarding land use planning – advocated, for example, by the European Environmental Agency (EEA 2003) and the EU (Council of the European Union 2009). There is thus a strong focus to increase community resilience through planning and governance, so that communities are prepared when a hazard strikes. Given this focus of policy development, a greater interest in *Guidance & Governance* responses may not be a bias of writers of articles away from the importance of

*Individual and Material Wellbeing* responses in Europe, but rather a manifestation of the EU's policy development and planning initiatives and an interest in how these are being implemented locally.

A follow-up matter is the question of why *Guidance & Governance* responses are more frequently reported from Europe than Northern America, given that both regions consist mostly of countries that rank high on the human development index





(HDI). Comparing cases from Europe and USA, Bubeck et al (2017) found that significantly lower standards for flood protection and damage mitigation policies apply in the USA even for some main coastal metropolises such as New York City and New Orleans. Indeed, flood risk management in the US is centered on flood insurance (Gouldby et al. 2017) in spite of the continued failure of the National Flood Insurance Program to achieve its objectives (Knowles & Kunreuther 2014). If we assume that the frequency of *Guidance & Governance* responses says something about the emphasis on policy and planning in a region, than we can conclude that Europe is more pro-active than similarly-developed Northern America, where there may thus be a need for more community-level responses of this class.

Looking next at responses of the class Awareness, we find that this is the least frequently used class of responses in Oceania and in Asia, with 8% of the community case studies from Oceania and 18% of Asian communities using responses of this kind. These are much lower frequencies compared to all other regions. Awareness responses aim to increase response capacity through improving understanding of hazards and better disseminating information about the threat conditions. Although it is possible that these kinds of responses are under-reported in studies conducted in these regions, our findings may also reflect a need for more efforts to increase hazard awareness in this region. In the case of Australia, this conclusion is in fact supported by the literature (e.g. Box et al. 2013; Sewell et al. 2016).

Finally, the use of *Relational Wellbeing* responses draws on collective action and the social fabric of the community for preparedness or protection against hazards. Overall, responses of this class are much less frequent (table 3), even though, for example, "community-based disaster preparedness" is gaining importance (Allen 2006) and emphasises the need to address social and political aspects of vulnerability (Allen 2006; Blaikie et al. 1994). Oceania is the region with the largest fraction (44% of all communities in our dataset) reporting community-level responses of the class *Relational Wellbeing*. Case studies discuss communities from Melanesia, Polynesia, Australia and New Zealand. Although it is not clear why relational wellbeing responses are more prominent in Oceania than in the other regions, this result does suggest that case studies from Oceania may provide a useful source of information to planners and decision makers who seek to enhance relational wellbeing in an effort to build social capital at the local level and to increase the resilience of their communities (Allen 2006).

### 3.3 Are there differences in responses to the different hazard types?

In order to answer this question, we examined whether the kinds of responses used differed between the types of hazards. Within each hazard type, *Individual and Material Wellbeing* responses are used most frequently, indicating that the overall result (for all hazard types together) applies to each hazard type individually (table 4). Comparing responses across hazard types, the fraction of communities that are reported to have employed *Relational Wellbeing* responses is highest in relation to climate hazards. Similarly, *Infrastructure* responses are most frequently reported in connection with water hazards, but




hardly ever in connection with climate hazards (2%). Responses from the class *Guidance and Governance* are more often reported for earth (45%) or water (43%) hazards than for weather (27%) or climate (25%). *Awareness* type responses are reported least frequently for weather hazards (15%).

### 3.4 Are communities using a multi-hazard approach?

Of the 318 communities in our data set, 53 (17%) are reported to have experienced hazard events from more than one hazard type (table 5). Although 17% is a surprisingly small fraction of the data, it does not imply that all other communities do not experience multiple hazards, but rather that the articles reviewed reported only one class of hazard. The largest number of communities with multi-hazard experiences is in Asia, the largest fraction in Africa. Most of the multi-hazard communities have experienced hazards of two types, and only a few communities in Asia and Latin America are reported to have

experienced three or four different hazard types.

There are increasing calls for comprehensive multi-hazard approaches to disaster risk, suggesting that response choices should be suitable for any kind of hazard (Council of the European Union 2009; PAHO 2011). To what extent are community-level hazard responses reflecting such a multi-hazard approach? Which classes of responses were most likely to be used across multiple types of hazard?

For this subset of 53 communities, reported to have experienced multiple hazards, were they likely to use a specific class of responses to deal with more than one type of hazard? Responses of the class *Individual and Material Wellbeing* are used by most (46) communities for more than one type of hazard; followed by the class *Relational Wellbeing*, used by 20 communities for more than one kind of hazard (table 6). All kinds of hazards have the potential to damage property and cut

people off from sources of food and income. When it comes to restoring people's basic needs and implementing actions to this effect, it makes little difference which kind of hazard caused the situation. Thus, it is not surprising that *Individual and Material Wellbeing* and *Relational Wellbeing* responses are the most frequent types of actions that are used across multiple hazards. The importance of *Relational Wellbeing* is surprising, however, because as shown in the previous section, only a relatively small proportion of communities within each region uses actions that fall in the class *Relational Wellbeing*. This

may imply that these kind of responses gain importance within a multi-hazard framework.

Responses of the three classes *Awareness*, *Guidance and Governance* and *Infrastructure* are less frequently used across multiple hazards, which suggests that these are more hazard specific. This certainly makes sense when we think of infrastructure responses, which aim to protect people and structures from the full effects of the hazard. The design of hazard resistant buildings and protective structures, such as sea walls or dams, requires knowledge about the hazard to anticipate

potential impact. Responses of the classes *Awareness* as well as *Guidance and Governance* aim to increase community preparedness disasters. Many elements of emergency preparedness are common to all hazards (WHO 2017), such as having to deal with chaos, the need for decision making in the context of uncertainties, the need for coordination and information management. The World Health Organisation Strategic framework for emergency preparedness recommends an all-hazards



approach for disaster preparedness that includes hazard-specific measures where necessary (WHO 2017). Nonetheless, the creation of hazard-specific response plans has been the global norm (PAHO 2011) and our results suggest that community-based responses to increase hazard preparedness are mostly hazard specific.

## 4. Conclusion

To our knowledge this study is the first comprehensive global overview of community-based responses to natural hazards. At the core was a typology of hazard responses, which emerged from comprehensive analysis of hundreds of real-world examples. The typology proved very useful for analysis of a global data set of community-level response actions by allowing a comparison of responses across several dimensions. Our results resonate with the findings of studies that are more specific in terms of geographic area or type of hazard. This helps to validate our approach, and facilitates comparison of the findings of local case studies, within a global context.

The typology presented here may be useful also for communities and community-focused agencies to structure their decision making and planning of response actions. A global overview of community response activities is also useful for national governments to aid decision making around hazard policies and how to best support and build local level response strategies, and for international aid agencies to see what kind of response actions need to be strengthen in different parts of the world. Our results have several implications for research and policy:

1. Within an extensive literature on hazards, there is relatively little empirical information in the context of communities and hazard responses. In particular there seems to be a dearth of empirical research on community-level hazard responses in Africa, where further case studies are needed to build a better picture of the actions that African communities are implementing in response to natural hazards.

2. Across the world, *Individual and Material Wellbeing* responses are the most frequently reported kinds of responses from every angle, and while this focus can be explained, there is something to be said for supporting *Relational Wellbeing* responses, recognizing the importance of building social capital and community resilience. Oceania is a leader on this front.

3. Specific regions of the world are lacking in their use of certain hazard responses classes. There may be various reasons for this, but to the extent that financial limitations are the cause, attention may be needed to rectifying the imbalance. There is a particular need, for example, to support hazard awareness activities in Oceania and Asia.

4. There is a lack of multi-hazard response approaches on the local level. Developing such strategies could help communities optimise their efforts.

5. Other regions might consider following Europe in its attention to *Guidance & Governance* – notably in developing integrative hazard response policies, that enable communities to utilise community-level institutions to develop preparedness approaches. In most developing regions, this would likely require suitable external funding support.





**Author contribution:**

Both authors contributed equally to the manuscript.

**Competing interests:**

The authors declare that they have no conflict of interest.

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





| Group | Class | Type |
|---|---|---|
| Human Wellbeing | Individual & Material Wellbeing | Food & Shelter |
| | | Financial |
| | | Livelihood Adaptation |
| | | Protect Belongings |
| | | Mobility |
| | | Psychological |
| | Relational Wellbeing | Community Fabric |
| Awareness & Governance | Awareness | Information & data |
| | | Knowledge |
| | Guidance & Governance | Governance |
| | | Management |
| | | Planning |
| Infrastructure | | Green Protection |
| | | Hard Protection |

Figure 1: Response examples were organised through a grounded theory aproach resulting in a typology of 15 types falling within 5 classes, within 3 aggregated groups of responses.





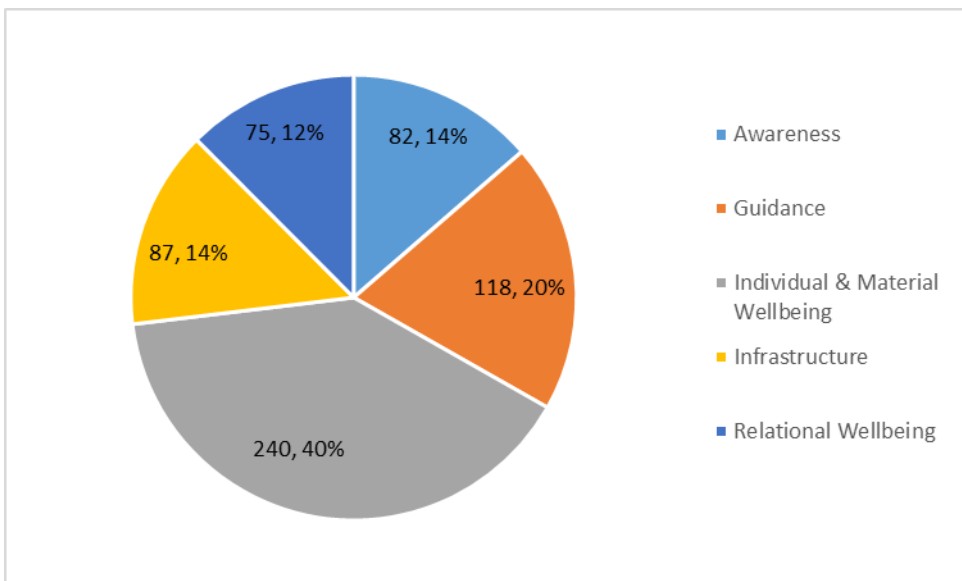

Figure 2: Frequency of use of response classes by communities; N= 602 communities utilise more than one type of response.





**Table 1: Date range and number of articles**

| Decade | 1981-1990 | 1991-2000 | 2001-2010 | 2011-2014 |
|---|---|---|---|---|
| Number of articles | 3 | 5 | 78 | 102 |

**Table 2: Spatial coverage of studies (188 articles). Research coverage represents the proportion of articles that address responses**
5   **in that region. The values exceed 100% because countries are covered by more than one study. Research coverage by region represents the proportion of countries within a region covered by the reviewed studies. (In other words, if all countries of a region were represented in our sample, coverage would be 100%.)**

| Region | Research coverage by articles (%) | Most researched country in region | | Research coverage by response examples (%) |
|---|---|---|---|---|
| Northern America | 26% | USA | 86% | 9% |
| Asia | 43% | India & Indonesia | 17% each | 42% |
| Europe | 12% | UK | 45% | 5% |
| Africa | 5% | South Africa & Sudan | 20% each | 6% |
| Oceania | 10% | Australia | 39% | 16% |
| Latin America and Caribbean | 15% | Mexico | 21% | 21% |

10   **Table 3: Fraction of all communities in each region, which are reported to have used responses from any of the five response classes. Totals exceed 1.00 because communities use responses from more than one class**

| World Region | Individual & Material Wellbeing | Relational Wellbeing | Awareness | Guidance | Infra-structure | Total |
|---|---|---|---|---|---|---|
| Africa | 0.92 | 0.33 | 0.42 | 0.00 | 0.33 | 2.00 |
| Asia | 0.75 | 0.20 | 0.18 | 0.32 | 0.20 | 1.65 |
| Europe | 0.39 | 0.26 | 0.43 | 0.61 | 0.39 | 2.09 |
| Latin America and the Caribbean | 0.60 | 0.25 | 0.44 | 0.44 | 0.52 | 2.25 |
| Northern America | 0.53 | 0.29 | 0.37 | 0.29 | 0.18 | 1.66 |
| Oceania | 0.72 | 0.44 | 0.08 | 0.31 | 0.47 | 2.03 |



**Table 4: Fraction of all communities for each hazard type, which have used responses from any of the five response classes. Totals exceed 1.00 because communities use responses from more than one class.**

| Hazard Type | Individual & Material Wellbeing | Relational Wellbeing | Awareness | Guidance | Infrastructure | Grand Total |
|---|---|---|---|---|---|---|
| Earth | 0.57 | 0.28 | 0.30 | 0.38 | 0.28 | 1.82 |
| Water | 0.70 | 0.26 | 0.27 | 0.36 | 0.42 | 2.03 |
| Weather | 0.84 | 0.27 | 0.15 | 0.27 | 0.24 | 1.78 |
| Climate | 0.79 | 0.36 | 0.26 | 0.25 | 0.02 | 1.68 |

**Table 5: Number of communities that are reported to have experienced more than one sub-type of hazard. Information was reviewed for 318 communities from five regions of the world.**

| World Region | Number communities | % multi Hazard | Multi hazard communities | 2 Hazard Subtypes | 3 Hazard Subtypes | 4 Hazard Subtypes |
|---|---|---|---|---|---|---|
| Africa | 12 | 33 | 4 | 4 | 0 | 0 |
| Asia | 157 | 17 | 27 | 23 | 3 | 0 |
| Europe | 23 | 13 | 3 | 3 | 0 | 0 |
| Latin America and the Caribbean | 52 | 15 | 8 | 5 | 3 | 1 |
| Northern America | 38 | 11 | 4 | 4 | 0 | 0 |
| Oceania | 36 | 19 | 7 | 7 | 0 | 0 |
| Total | 318 | 17 | 53 | 46 | 6 | 1 |

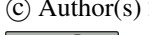



**Table 6: Number of communities exposed to multiple hazards that have utilised a specific class of responses .**

| Class of responses | Number communities using this class for multiple hazard groups | % of total number communities with multiple hazards |
|---|---|---|
| Individual & Material Wellbeing | 46 | 86.8 |
| Relational Wellbeing | 20 | 37.7 |
| Awareness | 10 | 18.9 |
| Guidance | 12 | 22.6 |
| Infrastructure | 10 | 18.9 |
| Number of communities exposed to multiple hazards | 53 | |