# Peer review of "A global comparison of community-based responses to natural hazards"

_Natural Hazards and Earth System Sciences, 2018_

## Referee Comment (RC1) · Anonymous Referee #1 · 28 Dec 2018

General comments The subject of the article 'A global comparison of community-based responses to natural hazards' is interesting and relevant to the scientific interests of NHESS. However, it seems that no conclusions can be drawn that can be generalized as the authors would like. The main problem is that they attempt to produce indicators of the communities' response to natural hazards based not on the results but on the frequency of previous studies per response category. But this way of analysis involves the risk of misleading conclusions, or at least of high uncertainty. Unless we accept that the trend of published studies on community response corresponds to the true trend of the communities' responses, the results of the article are not solid. Which is even more difficult to accept because the trend of publications is being examined globally and not for one region. But the number of publications on a topic and area
is composed of many parameters that can hardly be taken into account. The authors in some cases try to explain the number of publications by category and area taking into account relevant limitations, leaving the reader with the feeling that the frequency may not be significantly related to the actual trend. In that sense, I think that policy and governance implications may also be volatile. The methodology followed for the categorization of responses seems appropriate, even though different topics (adaptation/emergency/recovery responses to hazards) are altogether included in the analysis. I suggest the authors to reconsider their point of view, perhaps looking also at the temporal trend of the response priorities. That is to look at the change in the scientific interest, which could reflect a certain shift in response priorities through time. Or, to include other information in the aggregated Tables and the discussion section, e.g. ratios of the number of articles per number of the corresponding hazardous events for each world-region (EMDAT may have such data). This could show the low scientific interest or the low production of articles with such targets.

Specific comments The title refers to natural hazards; in the beginning of the Introduction, the reader assumes that the weather-related natural hazards will be addressed in the article, and particularly the ones threatening the coastal regions; in the beginning of Methods, geophysical hazards seem to be also included in the analysis. I suggest this to be clarified in the Intro and the abstract. It seems that the authors include in their analysis responses that correspond to 3 different timings with respect to the disaster: Before – during – after disaster responses, which mean: responses to prepare/adapt – emergency response – recovery response. These are 3 different topics and I would expect this to be addressed. Aren't they related also to different attitude of communities against natural hazards? Is this aspect important for the classification of responses? Other issues with respect to policy implications have not been addressed: in addition to positive aspects of the responses, did the writers in the review also distinguish negative aspects? eg, emergency responses that led to opposite results? In some cases it seems that conclusions may not reflect the tendency of the market. E.g. 3.3: green infrastructure for cities adaptation to climate change is however a growing sector. Could
the authors look at the temporal trends of response priorities?

Technical comments I understand that the number of articles reviewed is very large; could it be, however, provided within a Table, eg having 1 column for the categories, or the hazard type, or the area, and 1 for the references separated by ';'? I don't really understand Figure 1 and the percentages written. P8,I7: correction: in terms OF the total... P9,I6: correction: THEN we conclude...

---

## Author Comment (AC1) · 23 Jan 2019

Author response to review number 1 MS Title: A global comparison of community-based responses to natural hazards Author(s): Barbara Paterson and Anthony Charles MS No.: nhess-2018-353 MS Type: Research article

Response to Reviewer 1: Thank you for your review of our paper. We have answered each of your points below. Specific comments

1. The subject of the article 'A global comparison of community-based responses to natural hazards' is interesting and relevant to the scientific interests of NHESS [. . . ] it seems that no conclusions can be drawn that can be generalized as the authors would like. The main problem is that they attempt to produce indicators of the communities'

response to natural hazards based not on the results but on the frequency of previous studies per response category.

Response: We are not assessing "the communities' response to natural hazards" using "the frequency of previous studies." Instead, we examine the number of communities for which responses are reported across the global set of studies. This distinction may appear subtle but is indeed important. Using content analysis, we are recording descriptions of responses actions, which are then categorised into types and classes. Our data set consists of examples of response actions – each example has taken place in a specific community at a specific time. For each type or class, we then count how many communities have reported this kind of response action. Although the data is gleaned from published articles, we are not counting the number of articles but the number of communities for which response actions of a particular category have been reported. Note as well that we are not attempting to evaluate community responses in a normative sense, but rather we are grouping reported response actions to assess their frequencies, i.e. "what has been done". We agree with this reviewer that an evaluation of the success of the responses based on studies would be highly uncertain. We will need to provide a more detailed description of our methodology and the dataset to make this clearer.

2. But this way of analysis involves the risk of misleading conclusions, or at least of high uncertainty. Unless we accept that the trend of published studies on community response corresponds to the true trend of the communities' responses, the results of the article are not solid.

Which is even more difficult to accept because the trend of publications is being examined globally and not for one region. But the number of publications on a topic and area is composed of many parameters that can hardly be taken into account.

The authors in some cases try to explain the number of publications by category and area taking into account relevant limitations, leaving the reader with the feeling that the
frequency may not be significantly related to the actual trend.

Response: As we have explained above we are not suggesting that "the trend of published studies on community response corresponds to the true trend of the communities' responses". What we are suggesting is that the sample of community-based response actions that is gleaned from articles is a subset of all response actions. What may be contested is whether this sample is representative. We collected our data from peer reviewed articles. Since the scope of our study is intentionally global, there is no other feasible method of collecting the data, at least not to our knowledge. If the scope of the study was smaller, say regional or national, then it would be feasible to utilise additional or different data sources, such as grey literature or community surveys. But these sources/methods are not appropriate for a global overview. On the other hand, a global study with intensive resources and participation by all nations of the world could gather more extensive data, but this has not happened. Thus we engaged in an extensive comprehensive review and extracted response actions from the articles which we are using as proxies for what is going on in the world. However, as is the case with any review there is potential bias, research and publication is influenced by many factors as pointed out by the reviewer. Also some geographic areas e.g. Africa are under reported, which is also true for Emdat. We are focussing here on analytical categories that, allows investigation of hazard response actions in a similar way as Emdat enables analysis of disaster consequences. We appreciate the insightful comments by this reviewer. In our paper, we will need to add more detail to the methods section and also include these considerations into the conclusions section.

3. The methodology followed for the categorization of responses seems appropriate, even though different topics (adaptation/emergency/recovery responses to hazards) are altogether included in the analysis.

Response: We agree that the different phases of disaster response, i.e. adaptation, emergency, recovery are important analytical categories, but not appropriate for the purpose of our analysis. We explain this in more detail below.

4. I suggest the authors to reconsider their point of view, perhaps looking also at the temporal trend of the response priorities. That is to look at the change in the scientific interest, which could reflect a certain shift in response priorities through time.

Response: This is an interesting suggestion; although a time series analysis is beyond the scope of the present paper, we will consider this for a future analysis of our data.

5. Or, to include other information in the aggregated Tables and the discussion section, e.g. ratios of the number of articles per number of the corresponding hazardous events for each world-region (EMDAT may have such data). This could show the low scientific interest or the low production of articles with such targets. Response: Such a table could be included if it is felt that this would add value to the paper. But again, please note that the focus of this paper is on local communities and their hazard responses, not on the authors' scientific interest as such, or the research described in the articles, which are just the source of the data.

Specific comments

6. The title refers to natural hazards; in the beginning of the Introduction, the reader assumes that the weather-related natural hazards will be addressed in the article, and particularly the ones threatening the coastal regions; in the beginning of Methods, geophysical hazards seem to be also included in the analysis. I suggest this to be clarified in the Intro and the abstract.

Response: We are including earth hazards in our analysis and therefore need to change the sentence in the introduction to "Natural disasters such as floods, storms and earthquakes".

7. It seems that the authors include in their analysis responses that correspond to 3 different timings with respect to the disaster: Before – during – after disaster responses, which mean: responses to prepare/adapt– emergency response – recovery response. These are 3 different topics and I would expect this to be addressed. Aren't they

related also to different attitude of communities against natural hazards? Is this aspect important for the classification of responses?

Response: We agree with the reviewer that timing of responses is indeed an important analytical category. However, we found that this analysis is rather complex. We found that the categories before - during - after do not always correspond with the 3 phases preparation – emergency response – recovery. For instance, adaptation responses can be viewed as actions that communities implement based on past experiences with hazards and in anticipation of future events.

8. Other issues with respect to policy implications have not been addressed: in addition to positive aspects of the responses, did the writers in the review also distinguish negative aspects? eg, emergency responses that led to opposite results?

Response: As we explained above, we do not evaluate any of the responses, but simply record actions that have been carried out, irrespective of outcome. We found that only few studies mention unsuccessful response actions. We decided to nonetheless include these in our sample because we do not have the means to evaluate the success of response actions across all the studies from which we gathered data.

9. In some cases it seems that conclusions may not reflect the tendency of the market. E.g. 3.3: green infrastructure for cities adaptation to climate change is however a growing sector. Could the authors look at the temporal trends of response priorities?

Response: This is an interesting suggestion. However, we feel that market considerations would not add to the analysis we are reporting here, but would be a different study all together.

Technical comments

10. I understand that the number of articles reviewed is very large; could it be, however, provided within a Table, eg having 1 column for the categories, or the hazard type, or the area, and 1 for the references separated by ';'?

Response: We would be happy to provide this data. However, including all articles would inflate the references section. Such a table may perhaps be more appropriate as supplementary data.

11. I don't really understand Figure 1 and the percentages written.

Response: As explained above, for each class of responses we are counting how many communities have implemented response actions of this kind. The percentages reflect these relative frequencies.

12. P8,l7: correction: in terms OF the total. . . P9,l6: correction: THEN we conclude. . . Response: Thank you for these corrections – we will make the necessary edits.

---

## Referee Comment (RC2) · Anonymous Referee #2 · 28 Mar 2019

General comments: The article subject is interesting and fits the objectives of NHESS. However, some points of the article can lead to unwanted and unproved ideas. On the one hand, the use of only English literature, even if is scientific, can leave large areas of the world without a full coverage of hazards response by communities. From my point of view, you cannot be speaking about Africa with a low coverage when French is largely used by a high number of African countries. The same can happen with Latin America, where Spanish is used in a large amount of scientific papers. I think that a conclusion about a global overview is not absolutely correct, maybe a global overview of English hazards-related literature would be more appropiate. On the other hand, the date range of the articles used ranges from 1982 to 2014. However, it is said that the majority of research (102 papers) fall within the period 2001 to 2014, thus

leaving the starting period with only 86 articles. My question is if the starting period is represented at the same level of the final one? and if the results gathered from that period are as valuable as the ones from the final one? Could the difference explain a change within the scientific community about that topic? Finally, the conclusion is not clear enough regarding how the paper is useful to decision making and planning of response actions. I could find examples of actions and measures but I am not sure if that examples can help communities and policy-makers. Please note as well that the 4 key questions presented at the start of the paper are not clearly answered in the final section, maybe a short explanation could address this point. For instance, in point 3 you write about specific regions but they are not named, and in point 4 maybe a suggestion about the causes (if the articles used answer that questions) of the lack of multi-hazard responses could explain why that approach is missing around the world.

Specific comments: The authors use for the research a high number of articles but no reference is given to those papers, can be added a list including the references, with basic info such as authors, year of publication, title and affected area/hazard. It can be helpful to readers to undertand where and what are being studied and explained in the text. In page 10, when explaining the multi-hazard approach, a list of hazards could be added as the community answer can vary depending of the kind of hazards, including details about which are commonly related.

Technical comments: The authors should correct the references list. Some papers listed as a reference are missing from the text (Barker and McGregor, Sewell et al, WHO) or viceversa (Heijmans in p. 7, Hobson in p. 7, Knowles and Kunreuther in p. 9, Blaikie et al in p. 9) while others are cited as Gall or Gal. Finally, there are two references from Estaban et al, both published in 2013 which is not clear when one or another are used. Adding 2013a and 2013b should help the readers.

---

## Author Comment (AC3) · 10 Jun 2019

Author response to review number 2 MS Title: A global comparison of community-based responses to natural hazards Author(s): Barbara Paterson and Anthony CharlesMS No.: nhess-2018-353MS Type: Research article Response to Reviewer 2: Thank you for your review of our paper. We have answered each of your points below. General comments 1. The article subject is interesting and fits the objectives of NHESS. However, some points of the article can lead to unwanted and unproved ideas. On the one hand, the use of only English literature, even if is scientific, can leave large areas of the world without a full coverage of hazards response by communities. From my point of view, you cannot be speaking about Africa with a low coverage when French is largely used by a high number of African countries. The

same can happen with Latin America, where Spanish is used in a large amount of scientific papers. I think that a conclusion about a global overview is not absolutely correct, maybe a global overview of English hazards-related literature would be more appropriate.  Response: It is true that French is the official language in many African countries, and of course Spanish is spoken across Latin America. However, it is important to note that we are gleaning our data from peer review academic articles. It is widely accepted that English has become the language of choice for many international scholarly journals and the majority of non-non-English speaking scholars publish in English. In addition, many scholars are based at English speaking universities, irrespective of their own home language or the language spoken in their study area. Also, we would like to point out that under-reporting of disaster consequences in Africa was also found in EmDat data.

2. On the other hand, the date range of the articles used ranges from 1982 to 2014. However, it is said that the majority of research (102 papers) fall within the period 2001 to 2014, thus leaving the starting period with only 86 articles. My question is if the starting period is represented at the same level of the final one? and if the results gathered from that period are as valuable as the ones from the final one? Could the difference explain a change within the scientific community about that topic?

Response: A similar comment was made by reviewer 1. While a possible change in interest within the scientific community is not the focus of this paper, we will consider this for a future analysis of our data. We are not suggesting that the trend of published studies on community response corresponds to the true trend of the communities' responses. What we are suggesting is that the sample of community-based response actions that is gleaned from published, peer reviewed articles is a subset of all response actions. It may be contested whether this sample is representative, however, for a global analysis there is currently no other available data set. Âă 3. Finally, the conclusion is not clear enough regarding how the paper is useful to decision making and planning of response actions. I could find examples of actions and measures but I

am not sure if that examples can help communities and policy-makers.
 Response:

It is not our intention to suggest which kind of responses communities and policy makers should choose in any given situation. In fact, our data does not say which responses are best. We added a sentence in the second paragraph of the conclusion to clarify this point. If there is interest in the success of specific responses or the practicalities of any specific response approaches, then this would require individual communities or decision makers to follow up on the information provided here. But the point is that they would first need to know what is happening in other parts of the world. As we state in the conclusion, this paper provides information for communities and governments about what types of responses are happening around the world so that they can assess their own hazard planning by taking into account what others are doing. This is where the typology we developed provides a useful tool for practical use and for analysis, like that presented in this paper, which is based on the typology.

4. Please note as well that the 4 key questions presented at the start of the paper are not clearly answered in the final section, maybe a short explanation could address this point. For instance, in point 3 you write about specific regions but they are not named, and in point 4 maybe a suggestion about the causes (if the articles used answer that questions) of the lack of multi-hazard responses could explain why that approach is missing around the world.

Response: We have added a sentence about the key questions. As mentioned above our database does not include information on the reasons why particular responses were chosen in particular situations. Consequently, we cannot deduce the causes for the lack of a multi-hazard approach.

Specific comments 5. The authors use for the research a high number of articles but no reference is given to those papers, can be added a list including the references, with basic info such as authors, year of publication, title and affected area/hazard. It can be helpful to readers to understand where and what are being studied and explained in

the text.

Response: a similar comment was made by reviewer 1. We would be happy to provide this data. However, including all articles would inflate the references section. Such a table may perhaps be more appropriate as supplementary data.

6. In page 10, when explaining the multi-hazard approach, a list of hazards could be added as the community answer can vary depending of the kind of hazards, including details about which are commonly related. Âă Response: We agree that the community answer could vary depending on the kind of hazard, and included a sentence in the paragraph describing the multi hazard approach to highlight examples of the kind of hazards that might be addressed together using the same approach.

Technical comments

7. The authors should correct the references list. Some papers listed as a reference are missing from the text (Barker and McGregor, Sewell et al, WHO) or vice versa (Heijmans in p. 7, Hobson in p. 7, Knowles and Kunreuther in p. 9, Blaikie et al in p. 9) while others are cited as Gall or Gal. Finally, there are two references from Estaban et al, both published in 2013 which is not clear when one or another are used. Adding 2013a and 2013b should help the readers.

Response: We made all these corrections to the reference list as suggested.

Âă

---

## Referee Report (RR1)

**General comment**

The authors responded to all comments clearly. I found their arguments persuasive and therefore I believe that the article merits to be published.

**Minor change**

Both reviewers asked for a Table presenting the literature review and the authors suggested that a supplementary material could be provided. If this is feasible and the Editor also finds it necessary, I think that such Table could add to the scientific value/quality of the article.

---

## Author Response (AR2)

Barbara Paterson, PhD

Department of Environmental Science & Department of Finance, Information Systems, and Management Science

Sep 18, 2019

Dear Dr. Kotroni,

Please find attached the revision of our paper "a global comparison of community-based responses to natural hazards" for consideration for publication in *Natural Hazards and earth Systems Sciences*.

We completed the minor and technical revisions that were requested.

We feel that the resulting manuscript has benefitted much from the review process and hope that it is now fit for publication.

Yours sincerely,

Barbara Paterson